# Use of admission serum neutrophil gelatinase-associated lipocalin (NGAL) concentrations as a marker of sepsis and outcome in neonatal foals

**Malene Laurberg**[1¤a]*, **Claude Saegerman**[2], **Stine Jacobsen**[3], **Lise C. Berg**[3], **Sigrid Hyldahl Laursen**[3¤b], **Emma Hoeberg**[3¤c], **Elaine Alexandra Sånge**[3], **Gaby van Galen**[1,3]*

**1** Sydney School of Veterinary Science, University of Sydney, Camden, NSW, Australia, **2** Medicine and Surgery, Department of Veterinary Clinical Sciences, University of Copenhagen, Taastrup, Denmark, **3** Faculty of Veterinary Medicine, Research Unit in Epidemiology and Risk Analysis Applied to Veterinary Sciences (UREAR-ULiège), Fundamental and Applied Research for Animal and Health (FARAH) Center, University of Liege, Liege, Belgium

¤a Current address: HS Hestepraksis VetGruppen, Tarm, Denmark
¤b Current address: Equine Medical Consult, Hornbæk, Denmark
¤c Current address: Department of Companion Animal Clinical Sciences, Faculty of Veterinary Medicine, Norwegian University of Life Sciences, Ås, Norway
* gaby@equinespecialists.eu (GVG); malene.hshestepraksis@vetgruppen.dk (ML)

**Data Availability Statement:** All relevant data are within the paper and its Supporting information files.

## Abstract

### Background

Equine neonatal sepsis can be challenging to diagnose and prognosticate. Neutrophil gelatinase-associated lipocalin (NGAL), a new marker of renal damage and inflammation, can potentially be helpful.

### Objectives

To evaluate NGAL in neonatal foals with sepsis, and assess its relation to outcome.

### Animals

Foals $\leq$ 14 days, with admission blood analysis and stored serum.

### Methods

NGAL was measured on stored serum from 91 foals. Foals were scored for sepsis and survival and categorized according to sepsis status (septic, sick non-septic, healthy, and uncertain sepsis status) and outcome groups (survivors and non-survivors). The septic foals were further sub-categorized according to severity (normal sepsis, severe sepsis and septic shock). A Kruskal-Wallis test was used to compare serum NGAL concentrations in survivors and non-survivors, in the sepsis status groups, and in the sepsis severity groups. Optimal cut-off values for serum NGAL concentrations to diagnose sepsis and outcome were determined with receiver operating characteristic (ROC) curves. NGAL was compared to creatinine and SAA.

**Funding:** The author(s) received no specific funding for this work.

**Competing interests:** The authors have declared that no competing interests exist.

## Results

Median serum NGAL concentrations were significantly higher in septic than non-septic foals. However, serum NGAL concentrations did not differ between sepsis severity subgroups. Serum NGAL concentrations were significantly lower in survivors than in non-survivors. Optimal cut-off values of serum NGAL concentrations were 455 µg/L (sensitivity 71.4%, specificity 100%) and 1104 µg/L (sensitivity 39.3%, specificity 95.2%) for predicting sepsis and non-survival, respectively. NGAL correlated to SAA, but not to creatinine. NGAL performed similarly to SAA to diagnose sepsis.

## Conclusion

Serum NGAL concentrations may be useful for diagnosing sepsis and predicting outcome.

## Introduction

Neonatal sepsis is one of the most common and fatal conditions in neonatal foals [1], and sepsis therefore has a high welfare and economic impact [2]. Consequently, early diagnosis of sepsis with subsequent timely initiation of adequate treatment is of obvious benefit. However, sepsis is challenging to diagnose in early stages [3]. Blood culture is the preferred diagnostic test for sepsis, however it has low sensitivity [4], and results are only available after several days. Sepsis scoring [3] can provide early evidence, but several studies have indicated low sensitivity [5, 6]. To improve the diagnosis and prognostication in septic neonates, new or additional diagnostic approaches or diagnostic biomarkers are needed.

Neutrophil gelatinase-associated lipocalin (NGAL) is a relatively new biomarker of renal damage. In human medicine, plasma NGAL levels have also been shown to correspond with the severity of sepsis irrespective of the degree of renal dysfunction and to have prognostic ability, with higher serum concentrations in non-survivors than in survivors [7].

Previously, our research group has validated a porcine NGAL enzyme-linked immunosorbant assay (ELISA) for use on equine serum [8]. Further equine studies by our group have shown that an increase in NGAL is, similarly as in humans, indicative of renal injury in the horse [9], and that NGAL also responds strongly to inflammation [9]. Increased NGAL concentrations have been demonstrated in plasma, synovial fluid and/or peritoneal fluid in horses after experimental induction of systemic inflammation by intravenous lipopolysaccharide injection, following castration, and in horses with peritoneal or synovial inflammation and infection [9–11]. Currently, no information about NGAL is available for foals.

The objectives of this study were to determine whether serum NGAL concentrations can aid in 1) diagnosis of neonatal sepsis, and 2) predicting outcome in hospitalized neonatal foals.

## Materials and methods

### Study design

A retrospective study performed on clinical and analytical data on stored serum samples obtained from the large animal teaching hospital; University of Copenhagen; Taastrup Denmark.

## Ethics

Ethical approval was provided by the local ethical committee of the University of Copenhagen, Denmark. The need for owner consent was waived, and no consent was obtained.

## Case definition/study population

Foals presented between 2007 and 2017. Included foals were ≤ 14 days of age, had a hematology and biochemistry blood analysis performed on admission, and leftover serum that was stored at -20 °C.

## Recorded data

For all foals included in the study the following variables were recorded: signalment (age, breed, sex), history (duration of foaling, presence of dystocia, induction of foaling, presence of placentitis, mare's health status), clinical examination on admission (rectal temperature, heart rate, respiratory rate, presence of cold extremities, presence of petechial or scleral injection, presence of abnormal mentation, presence of infectious foci), laboratory findings on admission (leucocyte count, neutrophil count, band neutrophil count, presence of toxic neutrophils, SAA, L-lactate, fibrinogen, glucose, IgG concentration, and creatinine), final diagnosis and outcome. No further clinical data during hospitalization or after discharge was collected.

## Blood analysis

For all foals, blood was collected on admission by venepuncture using a vacutainer system into ethylenediaminetetraacetic acid (EDTA), serum and citrated tubes. Hematology and biochemistry analyses were performed on the ADVIA chemistry or hematology analyser (SIEMENS HEALTHCARE A/S, DENMARK). Serum amyloid A (SAA) analysis was performed with a Eiken LZ assay on the same ADVIA analyser. A serum sample was prepared for storage at -20˚C in cryotubes. Heterologous determination (duplicate analysis) of NGAL concentration in the serum samples was performed using NGAL ELISA (Kit no. 044, BioPorto Diagnostics, Hellerup, Denmark) that has previously been validated for this use with good assay accuracy [8]. Samples with concentrations above the calibration curve were further diluted to obtain a final NGAL concentration. Analysis was performed 2–12 years after sampling as recommended by the manufacturer.

## Scores

**Sepsis.**    A previously described sepsis scoring system [3] was used; this includes the history, clinical examination, rectal temperature, presence of infection sites, neutrophil count, band neutrophil count, morphology of neutrophils, glucose value, IgG value, fibrinogen value, presence of metabolic acidosis and the arterial oxygen concentration. Scores range from 0–16, with sepsis considered to be present at a score of ≥ 11. For each parameter with missing data both a maximum score and a minimum score was given (maximum score: parameters with missing data scored as high as possible based on the available clinical data; minimum score: parameters with missing data scored at 0). Foals thus received both a total maximum sepsis score and a total minimum sepsis score.

**Admission likelihood of survival.**    For assessing the likelihood of survival on admission, a scoring system was used that was developed by Dembek et al in 2014 [4], and recently validated in a large group of hospitalized foals up to 14 days of age [12]. This admission survival scoring system is based on the presence of prematurity, cold extremities, two or more

infectious or inflammatory sites, white blood cell counts, and blood glucose concentration. Scores range from 0 to 7 with a high score indicating high likelihood of survival.

## Categories

**Systemic inflammatory response syndrome (SIRS).** All foals were classified as having SIRS or not, following previously suggested age-appropriate SIRS criteria [13], Including rectal temperature, pulse, respiratory rate, WBC, band neutrophil count, blood glucose value and blood lactate. At least three of the before mentioned parameters needs to be abnormal, and at least one of them should either be abnormal rectal temperature or abnormal WBC to support a diagnosis of SIRS.

**Sepsis status.** The foals were divided into the following groups: septic, sick non-septic, healthy, and uncertain sepsis status. Foals classified as septic fulfilled one or more of the criteria as detailed in Table 1. Septic foals were further subdivided into normal sepsis, severe sepsis, and septic shock (definitions in Table 1). Foals classified as sick non-septic were diseased, but did not fulfill the criteria for sepsis, based on absence of the sepsis criteria and a minimal and maximal possible sepsis score both below 11. In absence of other parameters indicating sepsis, foals having a maximum possible sepsis score > 11 and a minimum possible sepsis score < 11 were classified as uncertain sepsis status.

**Outcome.** Foals were divided into groups based on survival to discharge: survivors and non-survivors. Foals euthanized due to financial restrictions were excluded from all statistical analyses concerning outcome.

## Data analysis

For numerical parameters mean, median, standard deviation (SD) and range were calculated. In a non-statistical manner (i.e. trend observation), serum NGAL and serum creatinine concentrations were described over the different groups, scores, and ages.

**Table 1. Definitions of sepsis and the sepsis subgroups.**

| Group | Definition |
|---|---|
| **Sepsis** | 1 or more of the following parameters:<br>A. A positive blood culture<br>B. Evidence of localized infection and presence of Systemic Inflammatory Response Syndrome (SIRS) [11]<br>C. A minimal and maximal possible sepsis score ≥ 11 [3][a]<br>D. The presence of multiple infection sites on necropsy. |
| **Sepsis severity groups** | |
| **Normal sepsis** | Sepsis without organ dysfunction, oliguria, hypotension or hypoperfusion, blood lactate < 2 mmol/L |
| **Severe sepsis** | Sepsis definition plus:<br>presence of organ dysfunction, hypotension or hypoperfusion, blood lactate > 2 mmol/L, oliguria or MAP < 60 mmHg [14] |
| **Septic shock** | Sepsis definition plus:<br>Persistent hypotension after fluid therapy or patients in need of/in treatment with dobutamine and/or vasopressors [14] |

MAP = Mean arterial blood pressure.

[a] When a foal had missing data for some parameters, the foal was given a maximum score (parameters with missing data scored as high as possible based on the available clinical data) and a minimum score (parameter scored at 0) for this specific parameter. This resulted in foals having a total maximum and minimum sepsis score.

## Statistical analysis

Statistical analyses were performed using STATA/SE 14.2 (StataCorp., College Station, Texas, USA). A *P* value< 0.05 was considered statistically significant.

Normality of the distribution of data was tested with Shapiro-wilk W test; data were found to be non-normal of distribution. A Kruskal-Wallis equality-of-population rank test was used to test the differences in serum NGAL concentrations, SAA and serum creatinine concentrations between groups or categories of foals. NGAL concentrations were compared to creatinine and SAA through linear correlation and a Spearman non-parametric test. Further comparisons of NGAL to creatinine concentrations and renal disease in this foal population are described in a different manuscript.

Receiver operating characteristic (ROC) curves were used to assess the diagnostic usefulness of serum NGAL concentrations for the prediction of neonatal sepsis and of outcome, and to identify optimal cut off values. For the ROC curve regarding sepsis status, septic *versus* non-septic and healthy foals were used, uncertain sepsis status foals were excluded. For the ROC curve regarding outcome survivors *versus* non-survivors were used [15]. Receiver operating characteristic (ROC) curves were also used to assess the diagnostic usefulness of serum SAA concentrations for the prediction of neonatal sepsis. For this ROC curve, septic *versus* non-septic and healthy foals were used, uncertain sepsis status foals were excluded.

## Results

### Study population

A total of 91 foals fulfilled the selection criteria. The study population consisted of ponies (4/91; 4.4%), trotters (7/91; 7.7%), thoroughbreds (3/91; 3.3%), warmbloods (52/91; 57.1%), and other breeds (24/91; 26.4%). The breed was not listed for 1 foal (1.1%). Sex was unknown for 5/91 (5.5%) cases, 38/91 (41.8%) were fillies, and 48/91 (52.7%) were colts. The age ranged from 1–12 days (median 2 days). Trend observation of NGAL concentrations in this population shows no apparent differences in these first 2 weeks of life.

### NGAL comparison to creatinine and SAA

There was no correlation between NGAL values and creatinine values in this population of foals (p = 0.76; Table 2). A statistically significant linear correlation was found between NGAL and SAA values (p < 0.001).

### NGAL as an aid to diagnose sepsis

There was an acceptable level of missing data for the sepsis score: an average of 9.9 out of the 14 parameters were obtained. Sixty-two point six percent of foals had available data for $\geq 10$ sepsis score parameters. Twenty-one foals (23.1%) were septic, 8 (11.4%) sick non-septic, and 62 (68.1%) had uncertain sepsis status (Table 2). Median serum NGAL concentrations in the septic group were significantly higher than in the sick non-septic and uncertain sepsis status groups (p = 0.004) There was little concentration overlap between sick non-septic foals and septic foals, but considerable overlap between septic foals and foals with uncertain sepsis status (Table 2 and Fig 1). No statistical difference was observed across the septic severity subgroups (in order to perform statistical comparison between sepsis severity subgroups, the severe sepsis group and the septic shock group had to be merged due to low case numbers) (p = 0.78) (Table 2). Median serum NGAL concentrations tended to increase with increasing sepsis scores (trend observation), but due to the sample size, data was not statistically analysed (Fig 2). The optimal cut-off value for serum NGAL concentrations predicting neonatal sepsis

**Table 2. Admission serum neutrophil gelatinase-associated lipocalin, serum creatinine and serum amyloid A concentrations in hospitalized neonatal foals based on sepsis categorization and outcome.**

| Foals (n = 91) | NGAL (µg/L) Mean ± SD; Median (Range) | Creatinine (mmol/L) Mean ± SD; Median (Range) | SAA (mg/L) Mean ±SD; Median (Range) |
|---|---|---|---|
| **Sick non-septic** (n = 8; 11.4%) | 200.0 ± 130.6; 160.2 (60.4–424.1) | 130.8 ± 90.3; 104 (68–347) | 29 ± 42.5; 14.8 (0.3–137.9) |
| **Uncertain sepsis status** (n = 62; 68.1%) | 429.0 ± 786.7; 180.9 (16.4–4973.0) | 128.6 ± 109.1; 91.5 (52–675) | 306.2 ± 451.1; 76 (0–2266.2) |
| **Septic** (n = 21; 23.1%) | 1125.7 ± 1223.9; 850.5 (28.6–4967.0) | 137.6 ± 112.7; 107 (35–516) | 1177.1 ± 1567.3; 496.1 (0–5996.7) |
| **Sepsis severity subdivision group: Normal sepsis** (n = 9; 42.9%) | 951.7 ± 670.9; 890.9 (37.8–2306.0) | 106.4 ± 78.6; 76 (49–298) | 1749 ± 2031.2; 567.4 (1.2–5996.7) |
| **Sepsis severity subdivision group: Severe sepsis** (n = 2; 9.5%) | 2623.1 ± 3314.8; 2623.1 (279.1–4967.0) | 129.5 ± 17.7; 129.5 (117–142) | 1193.6 ± 758.2; 1193.6 (435.4–1951.8) |
| **Sepsis severity subdivision group: Septic shock** (n = 10; 47.6%) | 982.8 ± 1075.4; 674.8 (28.6–3494.0) | 177.1 ± 138.4; 142 (35–264) | 659.1 ± 876.2; 314.4 (0–2819.7) |
| **Survivors** [*] (n = 63; 71.6%) | 359.4 ± 664.6; 177.8 (16.4–4973) | 122.1 ± 104.8 92 (51–516) | 314 ± 518.4; 52.4 (0–2819.7) |
| **Non-survivors** [*] (n = 25; 28.4%) | 1042.8 ± 1220.2; 661.3 (24.3–4967) | 145.1 ± 111.2; 107 (35–537) | 1024.4 ± 1456; 375.3 (0–5996.7) |

NGAL = neutrophil gelatinase-associated lipocalin, SAA = serum amyloid A.

[*]Three foals were excluded from this part of the study as they were euthanized due to financial restraint.

on admission was 455 µg/L, with a sensitivity and specificity of 71.4% and 100%, respectively (95% confidence interval; Fig 3 [A]).

There was no statistical difference in creatinine values between septic and non-septic foals (p = 0.87; Table 2). SAA was statistically different between septic and non-septic foals (p = 0.003). The optimal cut-off value for serum SAA concentrations predicting neonatal sepsis on admission was 62 mg/L, with a sensitivity and specificity of 85.7% and 87.5%, respectively (95% confidence interval; Fig 3 [B]).

## NGAL as a marker for the prediction of non-survival

The division of the foals over the outcome categories can be found in Table 2. Three foals were excluded following euthanasia caused by financial restraint. Of the 25 other non-survivors, 23 foals were euthanised and 2 died naturally. The reason for non-survival was clearly stated in the records for 40%, ranging from generalised sepsis to severe septic shock. For 60% of non-survivors the exact medical reason for non-survival was not clearly stated in the clinical records. No foals in this study population were euthanised for congenital abnormalities or malformations.

There were few foals with missing data for the survival score: an average of 5.9 out of 6 parameters were obtained. Ninety-eight point nine percent of foals had available data for ≥5 survival score parameters.

Median serum NGAL concentrations were significantly different between survivors and non-survivors (p-value = 0.0007; Table 2 and Fig 4), with concentrations overlapping in the

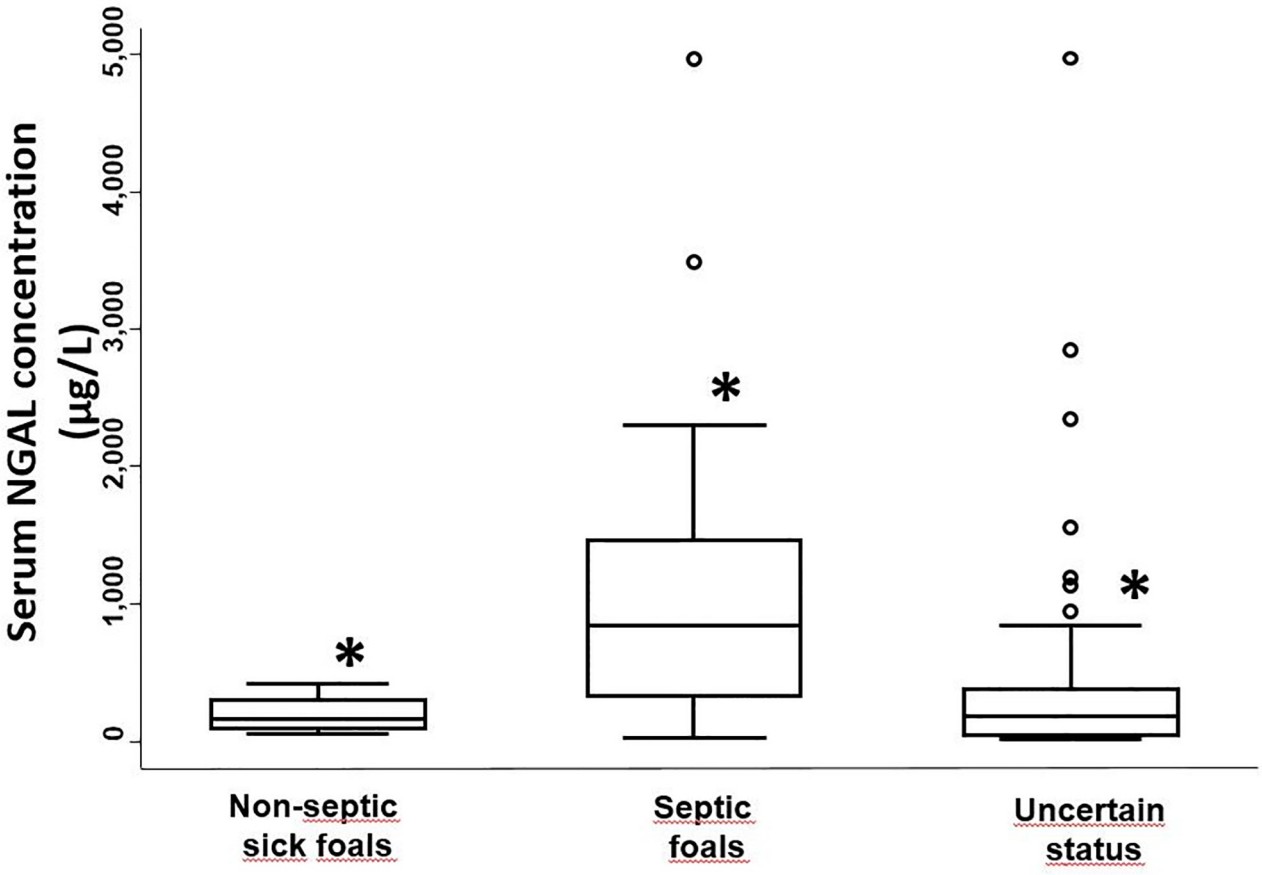

**Fig 1. Admission serum neutrophil gelatinase-associated lipocalin concentrations in hospitalized neonatal foals based on sepsis category.**
NGAL = neutrophil gelatinase-associated lipocalin. For each boxplot, the solid line in the rectangle represents the median value; the solid lines below and above each rectangle represent, respectively, the first and the third quartiles; adjacent lines to the whiskers represent the limits of the 95% confidence interval; small circles represent outside values. * = NGAL concentrations were significantly higher in septic foals compared to sick non-septic or uncertain sepsis status foals (p = 0.004).

two groups (Table 2). Serum NGAL concentrations decreased with increasing survival scores (trend observation; Fig 5). The optimal cut-off value for serum NGAL concentrations predicting non-survival was 1104 μg/L, with a sensitivity and specificity of 39.3% and 95.2%, respectively (95% confidence interval; Fig 6).

There was no statistical difference between serum creatinine values in survivors and non-survivors (p = 0.45; Table 2).

## Discussion

Serum NGAL concentrations have not been investigated in foals. Serum NGAL concentrations were significantly higher in septic than in non-septic foals and in non-surviving foals compared to those that survived, albeit with great concentration overlap in survivors and non-survivors. Using cut-off values of 455 μg/L and 1104 μg/L presence of sepsis and non-survival could be predicted with high specificity (100% and 95.2%, respectively). According to trend observation, serum NGAL concentrations increase with increasing sepsis score and decreasing survival scores. Moreover, NGAL did not show correlations with creatinine concentrations.

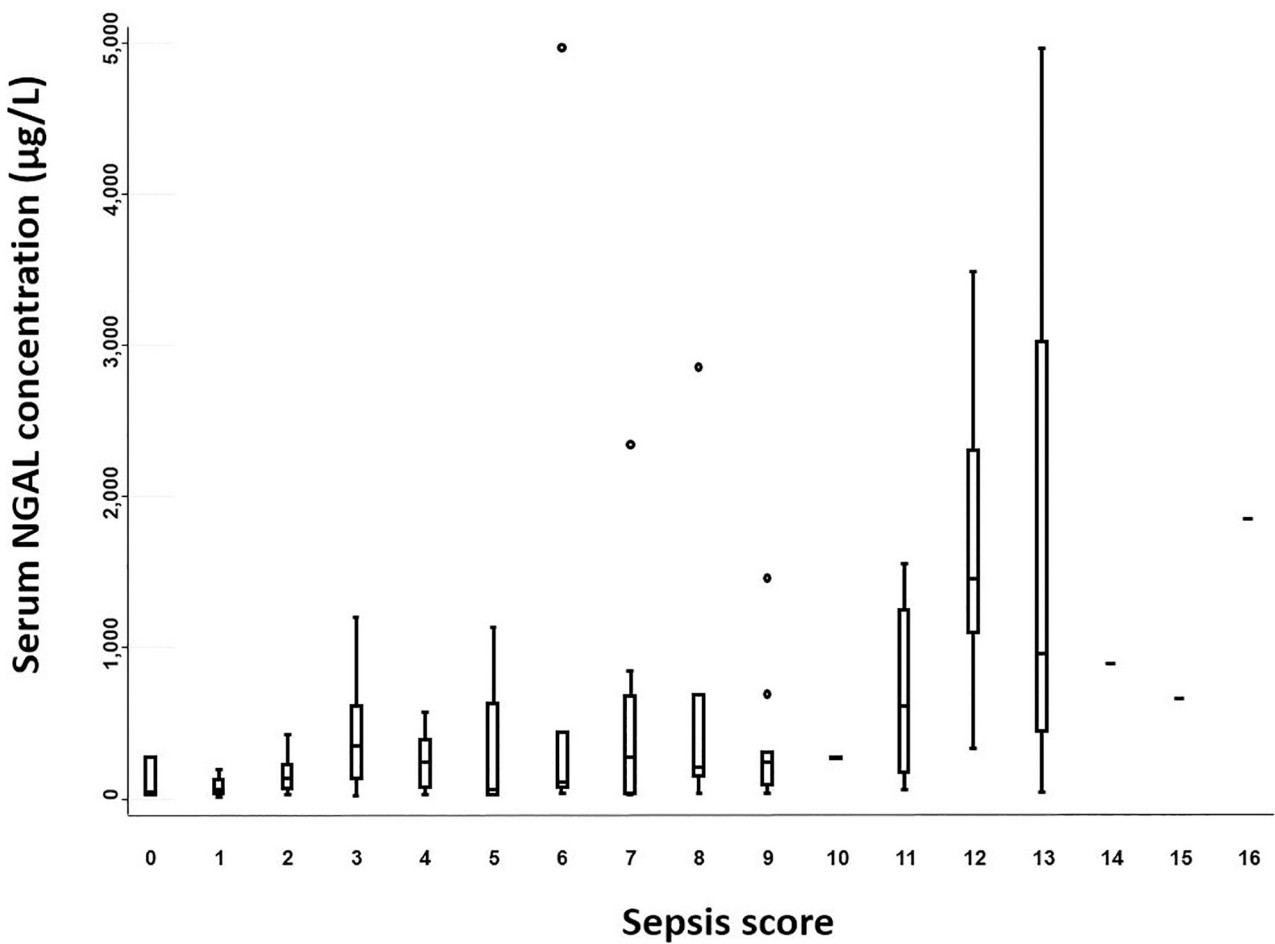

**Fig 2. Admission serum neutrophil gelatinase-associated lipocalin concentrations in hospitalized neonatal foals based on sepsis scores.**
NGAL = neutrophil gelatinase-associated lipocalin. For each boxplot, the solid line in the rectangle represents the median value; the solid lines below and above each rectangle represent, respectively, the first and the third quartiles; adjacent lines to the whiskers represent the limits of the 95% confidence interval; small circles represent outside values.

For serum SAA concentrations a cut-off of 62 mg/L could be used to predict the presence of sepsis with a sensitivity of 85.7% and specificity of 87.5%, and for NGAL a cut-off of 455 µg/L with a sensitivity of 71.4% and a specificity of 100%. To calculate these specificities of NGAL and SAA the cases with uncertain sepsis status were excluded. This resulted in low case numbers (21 septic foals versus 8 non septic foals) and thus wide confidence intervals (CI: 68.8 µg/L– 100 µg/L and CI: 47.4 mg/L– 99.7 mg/L respectively). Even with the wide confidence intervals, both NGAL and SAA performed within a similar range to predict the presence of sepsis. However, further studies with bigger case numbers are needed to better compare the usefulness of these two biomarkers to predict sepsis. A recent study with larger case numbers and similar sepsis categorisation revealed an optimal cut-off value of 1050 mg/L for SAA as predictor of sepsis [16], demonstrating how strong these values can fluctuate based on study population. Considering how useful and popular SAA has become in equine medicine, identifying another biomarker that performs similarly with regards to sensitivity and specificity during early testing phases is promising.

Assessment of serum NGAL could be a valuable adjunct to diagnosing and prognosticating sepsis in neonatal foals, and based on the high specificities, serum NGAL concentrations

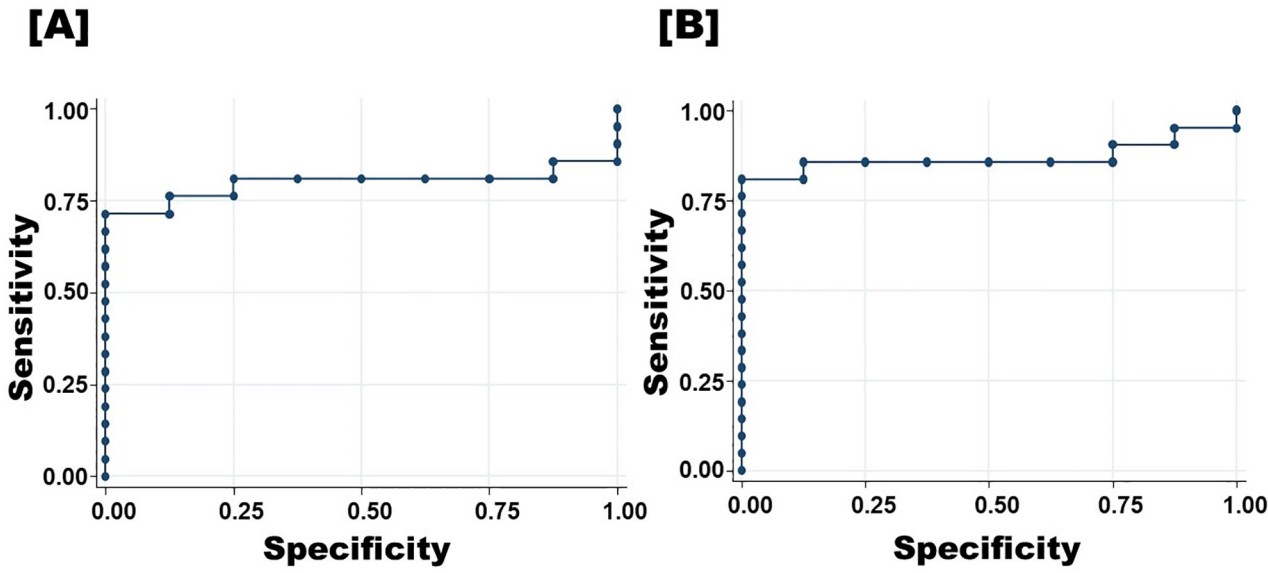

**Fig 3. Receiver operating characteristic curve of admission serum neutrophil gelatinase-associated lipocalin [A] and Serum amyloid A [B] concentrations in hospitalized neonatal foals to predict sepsis.** [A] Area under curve = 0.798 with se (area) = 0.0856; ━ fitted ROC curve with 95% CI; ● Observed values of the ROC curve. [B] Area under curve = 0.869 with se (area) = 0.0703; ━ fitted ROC curve with 95% CI; ● Observed values of the ROC curve.

would be most useful for ruling out sepsis, or non-survival. However, for severe diseases such as sepsis, which need early intervention for the patient to survive, it is desirable that the biomarker has high sensitivity. The lower sensitivity may have several potential explanations. Firstly, the gold standard diagnostics (sepsis scoring and blood culture) that NGAL was compared to have low sensitivities in themselves. Secondly, the sensitivity could reflect a limited diagnostic potential of NGAL, e.g. through increases in serum NGAL concentrations caused by concurrent or unrelated disease. Acute kidney injury is known to cause increased blood levels of NGAL [17, 18], and AKI may develop in foals with sepsis [19]. However, in septic human neonates, serum and urinary NGAL concentrations have been found to correlate strongly to the degree of systemic inflammation and not so much to the presence of AKI [20]. Also, the kinetics of the biomarker response may affect diagnostic and prognostic accuracy, especially in rapidly progressing diseases, such as sepsis. In adult horses, serum NGAL concentrations have been shown to peak at 16–36 hours after initial induction of endotoxemia [10], and foals could thus have been presented before serum NGAL concentrations increased maximally. Sensitivity can potentially be enhanced by combining NGAL with other diagnostics, such as the sepsis score, or by diagnostic tests that have a faster response time, such as leucocyte counts [21]. In this study, it was elected not to investigate how serum NGAL concentrations would perform integrated into the sepsis scoring. Based on the well-known difficulties in diagnosing sepsis in foals [5], it was decided to use the sepsis scoring to strengthen our sepsis (and hence case) definition, not allowing use of the sepsis score as a variable to be tested over the different groups. As a result of this more stringent approach to diagnosis, we had a large number of foals with uncertain sepsis status, thus further highlighting how difficult it is to diagnose sepsis in neonatal foals. Future studies should include serum NGAL concentrations in a sepsis scoring system to evaluate its effects on diagnostic accuracy. Another way to improve sensitivity would be to perform repeated or serial assessments, as shown for example for C-reactive protein, where repeated assessment within the first 24–48 hours of

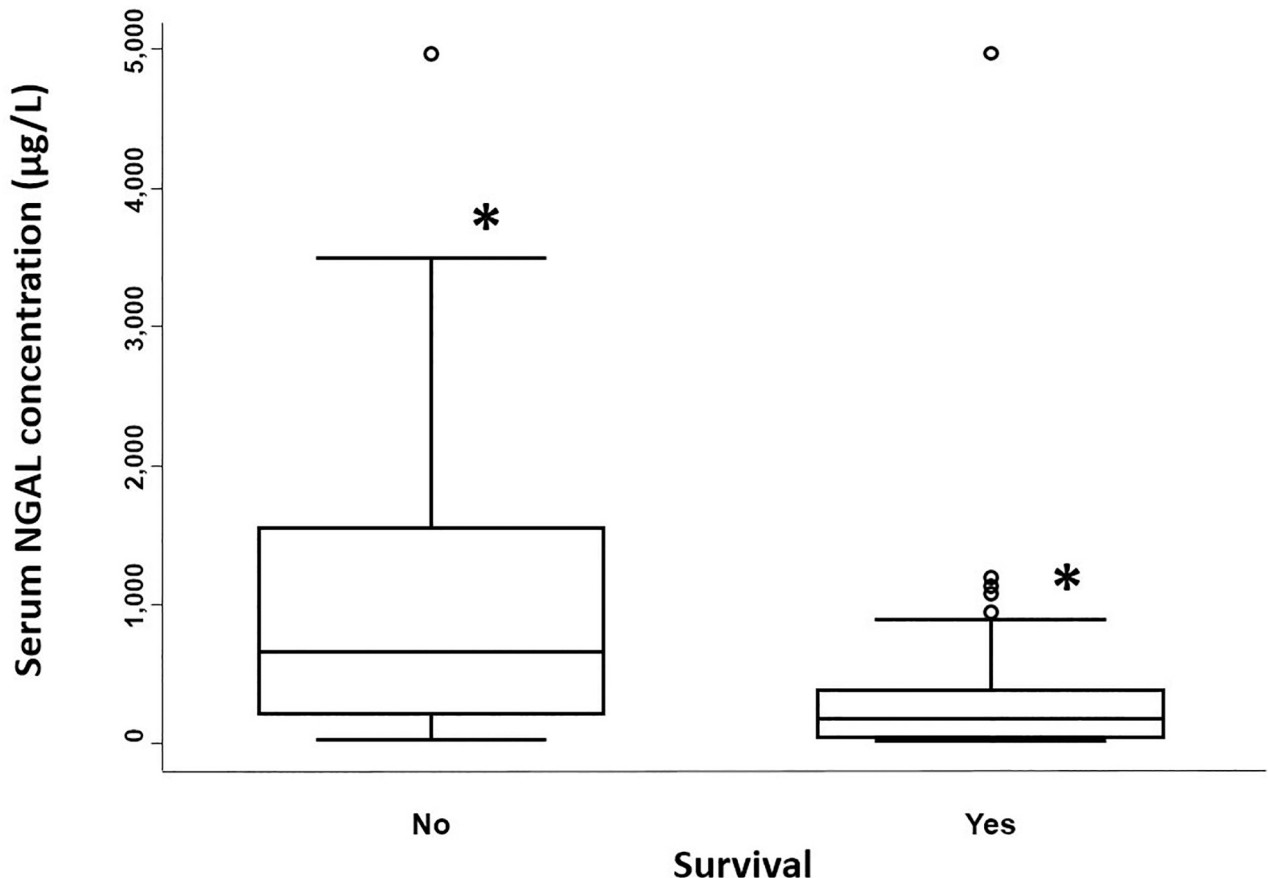

**Fig 4. Admission serum neutrophil gelatinase-associated lipocalin concentrations in hospitalized neonatal foals based on outcome.**
NGAL = neutrophil gelatinase-associated lipocalin. Three foals were excluded from this part of the study as they were euthanized due to financial restraint. For each boxplot, the solid line in the rectangle represents the median value; the solid lines below and above each rectangle represent, respectively, the first and the third quartiles; adjacent lines to the whiskers represent the limits of the 95% confidence interval; small circles represent outside values. * = significant difference in NGAL concentrations between survivors and non-survivors (p = 0.0007).

hospitalization in infants with sepsis improved sensitivity (20). But because of its high specificity, the absence of sepsis can be predicted with good confidence if the serum NGAL concentration in the foal is < 455 μg/L as false positives are rare.

More severe sepsis states were expected to have higher serum NGAL concentrations because of more extensive systemic inflammatory reaction, and organ dysfunction (such as developing AKI). This is also the case in human medicine, including paediatric medicine, where the more severe septic patient subgroups have higher NGAL values [7, 20–22]. However, the current study showed no difference between sepsis severity groups. This could be explained by low numbers in some of these subcategories, or by the fact that the most severe cases are potentially developing too fast and NGAL has not increased yet, or because of a compensatory anti-inflammatory response syndrome (CARS).

Using biomarkers to predict prognosis is highly desirable for several reasons, not least from an animal welfare point-of-view (sparing the animal the ongoing suffering when prognosis is hopeless) and for financial reasons (avoiding costly treatments for foals with grave prognosis). Our results suggest that serum NGAL concentrations can be included in decision-making, as they differed between survivors and non-survivors and were inversely associated with a survival score (lower serum NGAL concentrations with higher likelihood of survival; trend

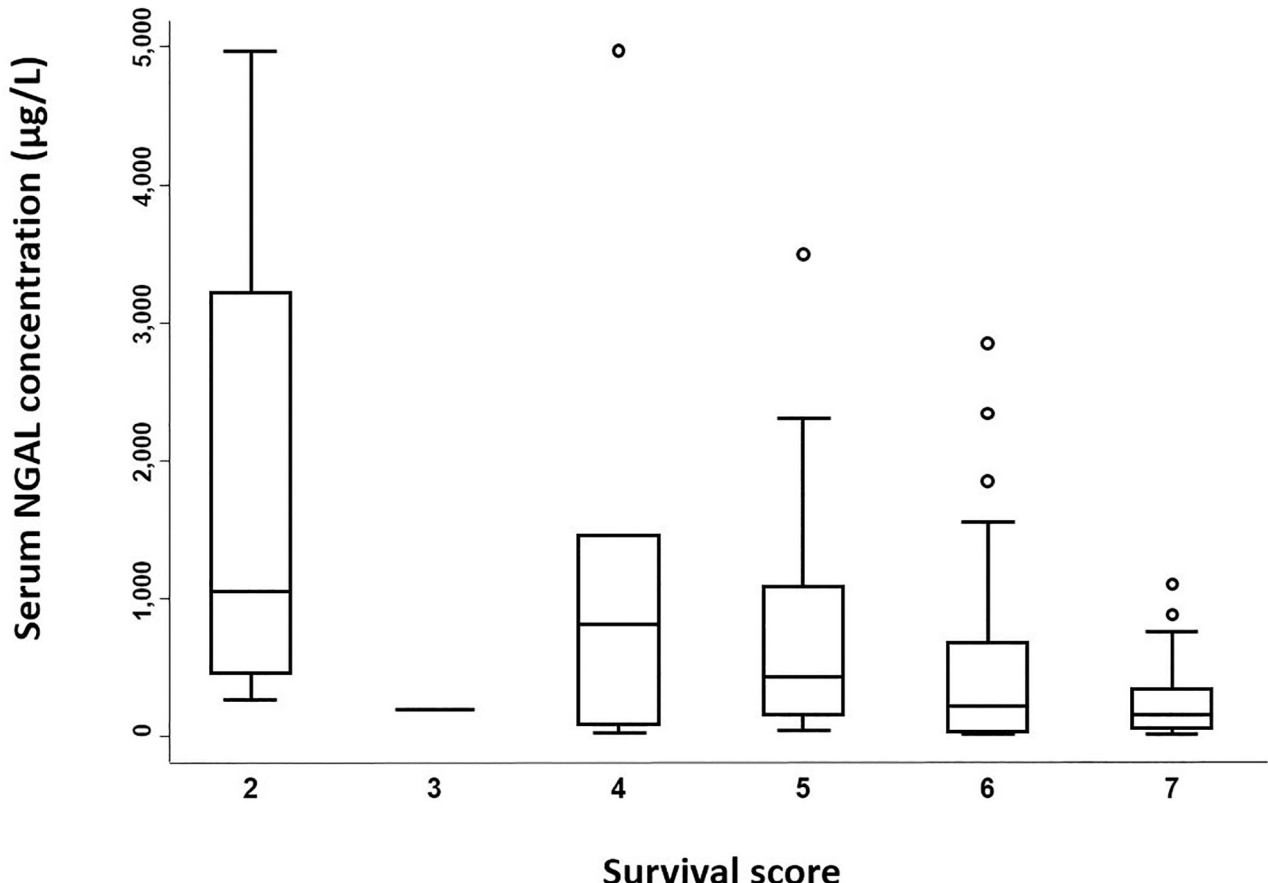

**Fig 5. Admission serum neutrophil gelatinase-associated lipocalin concentrations in hospitalized neonatal foals based on survival scores.**
NGAL = neutrophil gelatinase-associated lipocalin. Three foals were excluded from this part of the study as they were euthanized due to financial restraint. For each boxplot, the solid line in the rectangle represents the median value; the solid lines below and above each rectangle represent, respectively, the first and the third quartiles; adjacent lines to the whiskers represent the limits of the 95% confidence interval; small circles represent outside values.

observation). However, serum NGAL concentrations overlapped considerably between the two groups, with a cut-off of 1104 μg/L, with a low sensitivity of 39.3% and a specificity of 95.2%. This very low sensitivity and the overlap between groups limits its potential to be used as a prognostic marker in neonatal foals. In human medicine, contradicting data has been published on NGAL as prognostic marker: one study showed it to be a relatively robust predictor for 28 day survival in severely septic patients [23], while another study found limited utility for the prediction of survival [24]. Probably more even than in humans, outcome in foals can be determined by non or low inflammatory causes such as neonatal maladjustment syndromes or diseases that impact future possibility to participate in sport, such as musculoskeletal problems. Several of these conditions have no or limited effect on serum NGAL concentrations.

Serum NGAL concentrations did not correlate to serum creatinine in this study, and creatinine concentrations were not statistically different between groups, where serum NGAL concentrations were. This lack of correlation between NGAL and creatinine values was also found in human neonates [25] and in another study on adult horses by Siwińska et al. 2021 [26]. This could be explained by NGAL being a structural biomarker and creatinine being a functional biomarker of renal disease, but also by the fact that NGAL is known to also increase secondary to inflammation as shown in other studies [7, 9–11]. Furthermore, NGAL was correlated to

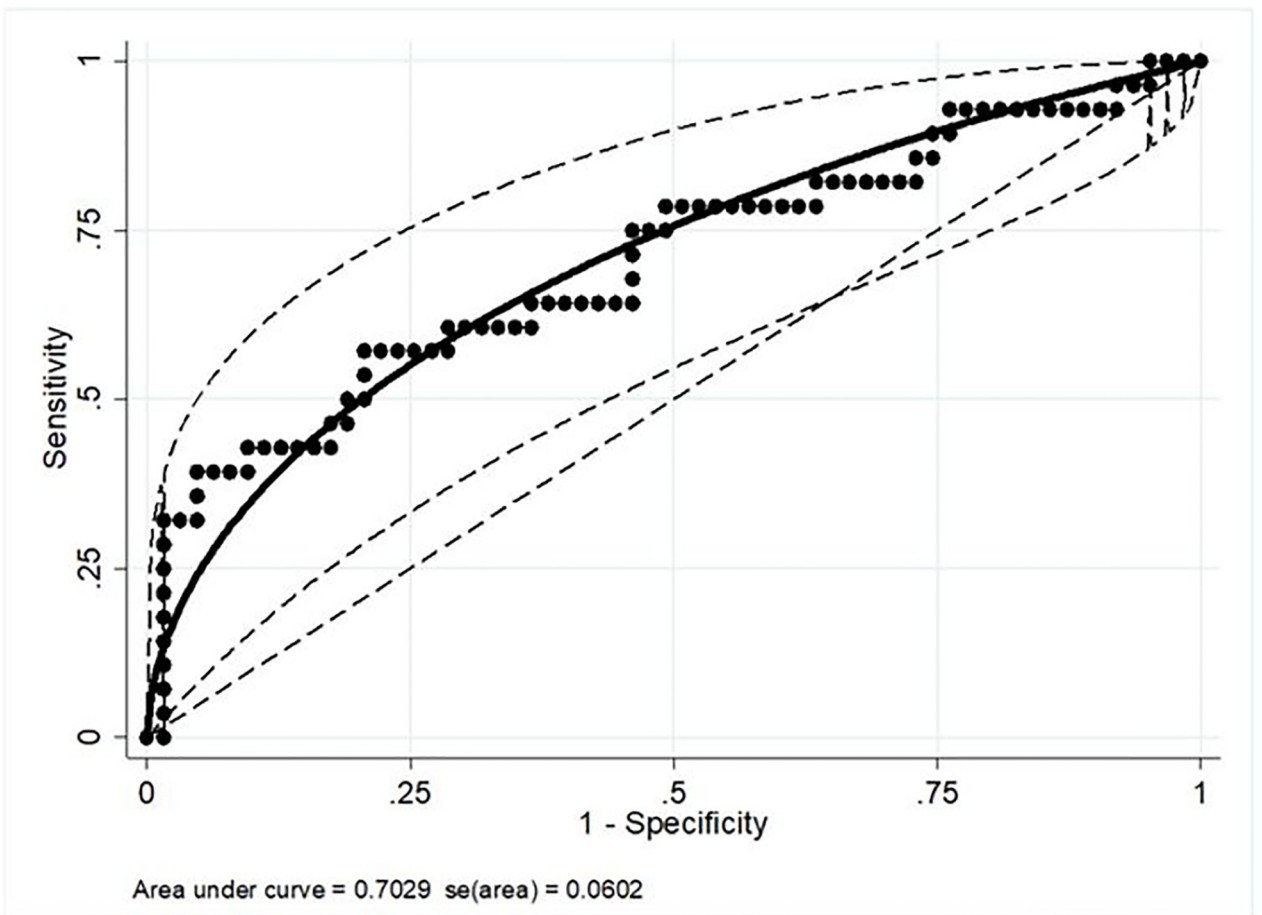

**Fig 6. Receiver operating characteristic curve of admission serum neutrophil gelatinase-associated lipocalin concentrations in hospitalized neonatal foals to predict non survival.** Area under curve = 0.7029 with se (area) = 0.0602; ━ fitted ROC curve with 95% CI; ● Observed values of the ROC curve.

SAA, which is an extensively used inflammatory marker for horses. These results further strengthen the hypothesis that independent of renal disease, NGAL is a marker for inflammation in horses; in this case for sepsis in neonatal foals. Also a statistically significant difference could be found in SAA concentrations between sepsis categories, which is in line with two previous studies on SAA in neonatal foals [16, 27].

Limitations of this study include the retrospective nature of the study, e.g. missing values and changes in management over time. However, a large portion of parameters and information needed for the study was available from the files, with relatively few missing data.

Very strict selection criteria were used for this study as we wanted to be as certain as possible on the presence of sepsis. Due to the small sample size this poses a limitation as most foals were then placed into the uncertain sepsis status category. As already mentioned above, the low sensitivity of the current gold standard diagnostics in diagnosing sepsis likewise may have affected the assessment of diagnostic accuracy of serum NGAL concentrations in our study through misclassification of foals. Furthermore, little is known about the normal reference values of serum NGAL concentrations in neonatal foals and its physiological behaviour in this age group. In this population and based on trend observation, there were no apparent

differences in NGAL concentrations during the first two weeks of life. However, whether age affects NGAL concentrations needs to be examined when reference ranges are established in a healthy population of foals. Currently, no NGAL bedside test is available for horses, limiting its potential to be used in clinical settings. A second study on the same population will describe the relation of NGAL with creatinine and urinary conditions in more depth.

In conclusion, serum NGAL concentrations increased with increasing sepsis score decreasing survival score, and differed between septic versus non-septic foals, and survivors versus non-survivors. Serum NGAL has a fair sensitivity, and a high specificity. When combined with other available diagnostic and prognostication tools for neonatal foals and keeping the kinetics of serum NGAL in mind, serum NGAL may be useful to assess this patient group. In future studies, diagnostic accuracy may be improved by combining NGAL with other markers or as an integral part of the sepsis score, or through use of repeated measurements. Studies directed at age-dependent reference values are also warranted.

## Supporting information

**S1 Data.**
(XLSX)

## Acknowledgments

All staff involved in the care of these foals is gratefully acknowledged.

## Author Contributions

**Conceptualization:** Stine Jacobsen, Lise C. Berg, Gaby van Galen.

**Data curation:** Stine Jacobsen, Sigrid Hyldahl Laursen, Emma Hoeberg, Elaine Alexandra Sånge.

**Formal analysis:** Malene Laurberg, Claude Saegerman, Lise C. Berg, Emma Hoeberg, Elaine Alexandra Sånge, Gaby van Galen.

**Funding acquisition:** Gaby van Galen.

**Investigation:** Malene Laurberg, Lise C. Berg, Emma Hoeberg, Elaine Alexandra Sånge, Gaby van Galen.

**Methodology:** Claude Saegerman, Stine Jacobsen, Lise C. Berg, Gaby van Galen.

**Project administration:** Gaby van Galen.

**Resources:** Sigrid Hyldahl Laursen, Gaby van Galen.

**Software:** Claude Saegerman.

**Supervision:** Stine Jacobsen, Lise C. Berg, Gaby van Galen.

**Validation:** Gaby van Galen.

**Visualization:** Malene Laurberg, Claude Saegerman, Gaby van Galen.

**Writing – original draft:** Malene Laurberg, Gaby van Galen.

**Writing – review & editing:** Malene Laurberg, Claude Saegerman, Stine Jacobsen, Lise C. Berg, Sigrid Hyldahl Laursen, Emma Hoeberg, Elaine Alexandra Sånge, Gaby van Galen.

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
