## [Decision Letter · Decision Letter 0]

18 Aug 2022

PONE-D-22-18141Use of serum neutrophil gelatinase-associated lipocalin (NGAL) concentrations to predict sepsis and outcome in hospitalized neonatal foals on admission.PLOS ONE

Dear Dr. van Galen,

Thank you for submitting your manuscript to PLOS ONE. After careful consideration, we feel that it has merit but does not fully meet PLOS ONE’s publication criteria as it currently stands. Therefore, we invite you to submit a revised version of the manuscript that addresses the points raised during the review process.

We look forward to receiving your revised manuscript.

Kind regards,

Angel Abuelo, DVM, MRes, MSc, PhD, DABVP (Dairy), DECBHM

Academic Editor

PLOS ONE

Journal Requirements:

2. Please note that PLOS ONE uses a single-blind peer review procedure. We would therefore be grateful if you could include in the information that has been redacted (line 75) for peer review in the manuscript.

3. In your Methods section, please provide additional details regarding participant consent from the owners of the animals. In the ethics statement in the Methods and online submission information, please ensure that you have specified (1) whether consent was informed and (2) what type you obtained (for instance, written or verbal). If the need for consent was waived by the ethics committee, please include this information.

5. We note that you have referenced (ie. Bewick et al. [5]) which has currently not yet been accepted for publication. Please remove this from your References and amend this to state in the body of your manuscript: (ie “Bewick et al. [Unpublished]”) as detailed online in our guide for authors

Reviewers' comments:

Reviewer's Responses to Questions

**Comments to the Author**

1. Is the manuscript technically sound, and do the data support the conclusions?

Reviewer #1: Partly

Reviewer #2: Yes

2. Has the statistical analysis been performed appropriately and rigorously? 

Reviewer #1: No

Reviewer #2: No

3. Have the authors made all data underlying the findings in their manuscript fully available?

Reviewer #1: Yes

Reviewer #2: Yes

4. Is the manuscript presented in an intelligible fashion and written in standard English?

Reviewer #1: Yes

Reviewer #2: Yes

5. Review Comments to the Author

Reviewer #1: The sample size of the study is rather small. Report confidence intervals for sensitivity and specificity and discuss power in discussion.

It seems based on the background information on other available methods, there isn’t much of a benefit to using this new biomarker? How do sensitivity and specificity compare? Discuss in methods why this method is worth pursuing further.

Objectives in the introduction don’t quite match the title. The title suggests you are predicting both sepsis and outcome.

What was the range and median of days from admission to discharge/death? And clarify there was no follow-up after discharge? (from my understanding)

Summarize the amount of missing data for the variables in the sepsis score calculation.

L55: what data exactly?

It states that statistical analyses were not conducted due to small sample size for comparing sepsis severity, but I recommend merging some of the smaller groups for purpose of comparison. For example, you could group severe sepsis with septic shock.

Add ROC curves so ranges of sensitivity and specificity can be seen.

L303-304: But age was not addressed in this study even though it was collected? I understand the low sample size, but reporting descriptive statistics stratified by the foal characteristics you collected would be a good baseline for future work to be based on.

Since NGAL has not been previously reported on in foals, it would be interesting to summarize NGAL values by the other data collected. If there are correlations with other biomarkers and if NGAL is different based on other characteristics (including age).

The 3rd figure is missing a label for x axis.

Reviewer #2: General comments: This is a well-written manuscript describing differences in serum neutrophil gelatinase-associated lipocalin (NGAL) concentrations between septic and non-septic foals, as well as between surviving and non-surviving foals. As the authors acknowledge, availability of a serum biomarker that could predict both sepsis and survival would be useful to equine clinicians. Unfortunately, although NGAL concentrations were specific, they had relatively low sensitivity for either sepsis or survival. Consequently, it remains unclear whether NGAL will become a practical biomarker, as compared to currently used clinical and laboratory parameters used to support sepsis and to predict outcome. The authors acknowledge this problem in their discussion of limitations of the manuscript as they state that: “In future studies, diagnostic accuracy may be improved by combining NGAL with other markers or as an integral part of the sepsis score”. This statement also points out one of this reviewer’s main concerns with this manuscript: the authors have collected clinical and laboratory data from this cohort of foals, yet they have not made any attempt to compare NGAL to other clinical or laboratory data. The manuscript could be much stronger if this type of comparison would be made though multivariable linear regression analysis of the data collected; specifically, well-documented risk factors for sepsis (failure of passive transfer) or simple laboratory parameters that would support sepsis (leucocyte count, glucose concentration, others). As an example, the authors are referred to: Giguère S, Weber EJ, Sanchez LC. Factors associated with outcome and gradual improvement in survival over time in 1065 equine neonates admitted to an intensive care unit. Equine Vet J. 2017 Jan;49(1):45-50 – at the least this manuscript should be referenced in this manuscript. Next, it would be useful to have more information about the non-surviving foals: 1) how many died/were euthanized; 2) if euthanized – due to generalized sepsis (non-responsive flat foal) or localized sepsis (e.g., musculoskeletal system); 3) were any euthanized for congenital anomalies (would not be suspected to have increased NGAL); and 4) age at admission of survivors vs non-survivors (a previously documented risk factor). Further, if a foal was euthanized for localized musculoskeletal sepsis at 2-3 weeks of age but was otherwise healthy, NGAL concentrations might be expected to be lower than in 1-3-day-old flat foal with generalized sepsis. Another major limitation of this manuscript is the inability to discriminate between septic and non-septic foals as the largest group (n=62) was the uncertain sepsis status group. This problem could be considered a “fatal flaw’” of the study. Finally, since NGAL is also considered a biomarker of acute kidney injury (AKI), it would be of interest to compare NGAL concentrations to creatinine concentrations in these groups of foals. Although I will leave the decision on acceptability of the manuscript as written to the Editor, I think this manuscript would be considerably strengthened if these points of concern would be addressed. As written, this manuscript would not provide much impetus for a commercial agency to attempt to develop a foal-side NGAL ELISA test.

There are also a few minor edits that should be addressed that are detailed below under specific comments should the authors pursue a further revision.

Specific comments:

Introduction

Line 70 – delete “the” before diagnosis

Materials and Methods

Lines 96-98 – What is “heterologous determination”? Was NGAL measured in duplicate/triplicate? Was there any internal quality control samples for NGAL to determine variance in the clinical use of the assay?

Results

Table 1. It would be of interest to include (n=#) in Table 1 for the initial 4 lines on how sepsis was documented – realizing that some foals would have more than one criterion to support sepsis; could then possibly eliminate lines 171-172

Line 177 – suggest changing “seemed” to “trended”

Discussion

Line 226 – suggest changing “have never previously been” to “not been”

Lines 243-245 – again, the authors could examine the potential relation between AKI and sepsis in this group of foals

Line 247 – it would be useful to define the nature of the “insult” as the proceedings cited [8] may not be widely available to treaders of this manuscript

Lines 249-251 – the authors are stating the obvious – again raising the question why a multivariable approach was not pursued in this study

Line 257 – consider changing “capacity” to “accuracy”

Line 260 – delete “significantly” – unnecessary

Line 268 – change “can” to “could”

Lines 275-276 – “differed between survivors and non-survivors and were inversely associated

with a survival score (lower serum NGAL concentrations with higher likelihood of survival” – this statement suggests a correlation analysis was performed – but there is no description of such an analysis in the Methods

References

Double check to confirm accuracy and consistency of presenting the references

Figures – should include some notation (asterisks or other marker) to indicate significant differences

6. PLOS authors have the option to publish the peer review history of their article (what does this mean?). If published, this will include your full peer review and any attached files.

Reviewer #1: No

Reviewer #2: **Yes: **Harold C. Schott II

---

## [Decision Letter · Decision Letter 1]

7 Dec 2022

PONE-D-22-18141R1Use of serum neutrophil gelatinase-associated lipocalin (NGAL) concentrations as a marker for sepsis and outcome in hospitalized neonatal foals on admission.PLOS ONE

Dear Dr. van Galen,

Thank you for submitting your manuscript to PLOS ONE. After careful consideration, we feel that it has merit but does not fully meet PLOS ONE’s publication criteria as it currently stands. Therefore, we invite you to submit a revised version of the manuscript that addresses the points raised during the review process.

We look forward to receiving your revised manuscript.

Kind regards,

Angel Abuelo, DVM, MRes, MSc, PhD, DABVP (Dairy), DECBHM

Academic Editor

PLOS ONE

Journal Requirements:

Reviewers' comments:

Reviewer's Responses to Questions

**Comments to the Author**

1. If the authors have adequately addressed your comments raised in a previous round of review and you feel that this manuscript is now acceptable for publication, you may indicate that here to bypass the “Comments to the Author” section, enter your conflict of interest statement in the “Confidential to Editor” section, and submit your "Accept" recommendation.

Reviewer #2: (No Response)

2. Is the manuscript technically sound, and do the data support the conclusions?

Reviewer #2: Yes

3. Has the statistical analysis been performed appropriately and rigorously? 

Reviewer #2: No

4. Have the authors made all data underlying the findings in their manuscript fully available?

Reviewer #2: Yes

5. Is the manuscript presented in an intelligible fashion and written in standard English?

Reviewer #2: Yes

6. Review Comments to the Author

Reviewer #2: General comments: This is a well-written revised manuscript describing differences in serum neutrophil gelatinase-associated lipocalin (NGAL) concentrations between septic and non-septic foals, as well as between surviving and non-surviving foals. As the authors acknowledge, availability of a serum biomarker that could predict both sepsis and survival would be useful to equine clinicians. Unfortunately, although NGAL concentrations were specific, they had relatively low sensitivity for either sepsis or survival. Consequently, it remains unclear whether NGAL will become a practical biomarker, as compared to currently used clinical and laboratory parameters such as glucose and IgG concentrations to support sepsis and to predict outcome.

In their revised manuscript, the authors have addressed some of the concerns raised in my initial review of this manuscript. Importantly, they have added information about serum creatinine concentrations. Unfortunately, they failed to perform a more robust analysis of the data collected as suggested in my initial review. Rather, they refer to data being presented in another manuscript in preparation. Again, this is disappointing and makes the usefulness of this report questionable. At the least, it would have been useful to include information about current “biomarkers” of sepsis such as serum glucose and IgG concentrations on admission. If NGAL concentration performed no better than these already accepted biomarkers of sepsis and survival, there would seem to be little incentive to further investigate NGAL (not just because we can measure it). Next, the validity of NGAL measurements warrants a bit more discussion in methods and discussion. In the authors validation study, linearity was only assessed by dilutional parallelism to concentrations up to 600 µg/L (reference 8, Figure 1) and it appears there were only 2 samples with values of 2000-3000 µg/L (reference 8, Figure 3). However, in the current manuscript, NGAL values approaching 5000 µg/L are presented in the Figures – how do we know if these values are accurate? It would be useful for the authors to present a “range” for accurate assay performance in the Methods section of this manuscript as well as mentioning potential assay limitations in the Discussion. This point is important as values at the extreme ends of the data set can have a substantial influence on correlations and development of ROCs.

Further, the authors suggest that one reason sensitivity might have been low could have been that admission NGAL concentrations had not reached maximal values in earlier stages of sepsis. However, the same limitation could also used for diagnostic accuracy. If foals had a low NGAL at admission, that subsequently increased as sepsis progressed, the “low” value at admission would suggest a high likelihood of survival that would have changed as sepsis progressed. This limitation should also be acknowledged in the Discussion.

Once again, the authors state in their conclusion: “In future studies, diagnostic accuracy may be improved by combining NGAL with other markers or as an integral part of the sepsis score, or through use of repeated measurements.” It is a shame that they did not make the effort to perform these analyses in this manuscript since all the data was available to them.

Consequently, the only novel aspect to this manuscript is that it is the first to report serum NGAL concentrations in foals. As such, this data is worthy of publication but this reviewer again remains disappointed it the lack of further analyses that could have been made with this data set

There are also a number of minor edits that should be addressed that are detailed below under specific comments should the authors pursue a further revision.

Specific comments:

Title: suggest rewording to: Use of admission serum neutrophil gelatinase-associated lipocalin (NGAL) concentrations as a marker of sepsis and outcome in neonatal foals

Abstract

Line 25 – delete “for”

Line 33 – suggest using another term for “normal” sepsis – a bit of an oxymoron as sepsis is certainly not normal

Introduction

Line 48 - delete “most” before fatal

Line 53 – suggest replacing “results” with “evidence”

Lines 56-60 – combine sentences into a single paragraph

Line 62 – change “for use in horses” to “for equine serum”

Lines 68-69 - suggest changing “Knowledge of equine NGAL is sparse, and no information on NGAL in foals is currently available.“ to “Currently, no information about NGAL is available for foals.”

Materials and Methods

Line 99 – can delete “standardly”

Line 123 – add “concentration” after glucose

Line 129 – change “(13). Including” to “(13), including” – otherwise sentence starting with “Including” is an incomplete sentence

Line 132 – add “to support a diagnosis of SIRS” to the end of the sentence

Line 135 – change “as explained” to “detailed”

Line 136 – again, change “normal” sepsis; also in Tables 1 and 2

Line 166 – “Spaerman” should be “Spearman”

Results

Line 196 – suggest changing “trended” to “tended”

Table 2. It would be preferable to list data as mean ± SD, followed by median (range)

Discussion

Line 292 – add a period after (19)

Line 293 – consider changing “capacity” to “accuracy”

Line 296 – consider changing “sufficiently” to “maximally”

Line 348 – consider changing “selected” to “used” to limit repetition

References

Double check to confirm accuracy and consistency of presenting the references. Reference 3 has authors names in ALL CAPITAL LETTERS. Reference 25 has CAPITAL letters starting most words in the manuscript title

Figures – should include some notation (asterisks or other marker) to indicate significant differences

7. PLOS authors have the option to publish the peer review history of their article (what does this mean?). If published, this will include your full peer review and any attached files.

Reviewer #2: No

---

## [Decision Letter · Decision Letter 2]

20 Mar 2023

PONE-D-22-18141R2Use of admission serum neutrophil gelatinase-associated lipocalin (NGAL) concentrations as a marker of sepsis and outcome in neonatal foals.PLOS ONE

Dear Dr. van Galen,

Thank you for submitting your manuscript to PLOS ONE. After careful consideration, we feel that it has merit but does not fully meet PLOS ONE’s publication criteria as it currently stands. Therefore, we invite you to submit a revised version of the manuscript that addresses the points raised during the review process. Reviewer #2 still has some additional queries that are relevant. Particularly, their suggestion of conducting further analyses on the SAA data would strengthen the current manuscript and should not take long to perform. I am aware of the need for a timely publication but feel that the reviewer's suggestion is worthwhile and would result in a more robust and useful manuscript.

We look forward to receiving your revised manuscript.

Kind regards,

Angel Abuelo, DVM, MRes, MSc, PhD, DABVP (Dairy), DECBHM

Academic Editor

PLOS ONE

Journal Requirements:

Reviewers' comments:

Reviewer's Responses to Questions

**Comments to the Author**

1. If the authors have adequately addressed your comments raised in a previous round of review and you feel that this manuscript is now acceptable for publication, you may indicate that here to bypass the “Comments to the Author” section, enter your conflict of interest statement in the “Confidential to Editor” section, and submit your "Accept" recommendation.

Reviewer #2: (No Response)

2. Is the manuscript technically sound, and do the data support the conclusions?

Reviewer #2: Yes

3. Has the statistical analysis been performed appropriately and rigorously? 

Reviewer #2: Yes

4. Have the authors made all data underlying the findings in their manuscript fully available?

Reviewer #2: Yes

5. Is the manuscript presented in an intelligible fashion and written in standard English?

Reviewer #2: Yes

6. Review Comments to the Author

Reviewer #2: General comments: This well-written second revision addresses most of the concerns raised in my review of the first revision. However, they have not revised “normal sepsis” - an oxymoron. I suggest another categorization such as mild sepsis, although admittedly “mild” sepsis would also be challenging to define. I will let the Editor make a final decision on this concern.

Next, in response to my previous review, the authors did compare NGAL to other laboratory measures of inflammation – thanks for doing this. However, addition of SAA to the manuscript has raised another question. Specifically, was sensitivity and specificity of admission SAA concentration for sepsis, sepsis severity, and outcome better than NGAL. Since SAA can be measured stall-side, this would be an important comparison that could be included in this manuscript. Again, I do not want to further prolong review of this manuscript so I will defer to the Editor as to whether this additional analysis should be performed and included in a final revision of this manuscript.

Specific comments:

Abstract

Line 33 – suggest using another term for “normal” sepsis – a bit of an oxymoron as sepsis is certainly not normal

Introduction

Line 63 – suggest changing “of” to “by” at end of line

Materials and Methods

Lines 95-96 – need to add SAA here as well

Line 132 – “Band” should be “band”

Line 139 – again, change “normal” sepsis; also in Tables 1 and 2

Discussion

Line 305 – add a period after (19)

7. PLOS authors have the option to publish the peer review history of their article (what does this mean?). If published, this will include your full peer review and any attached files.

Reviewer #2: No

---

## [Editor Report · Decision Letter 3]

3 May 2023

Use of admission serum neutrophil gelatinase-associated lipocalin (NGAL) concentrations as a marker of sepsis and outcome in neonatal foals.

PONE-D-22-18141R3

Dear Dr. van Galen,

We’re pleased to inform you that your manuscript has been judged scientifically suitable for publication and will be formally accepted for publication once it meets all outstanding technical requirements.

Kind regards,

Angel Abuelo, DVM, MRes, MSc, PhD, DABVP (Dairy), DECBHM

Academic Editor

PLOS ONE
---

## [Editor Report · Acceptance letter]

9 May 2023

PONE-D-22-18141R3 

Use of admission serum neutrophil gelatinase-associated lipocalin (NGAL) concentrations as a marker of sepsis and outcome in neonatal foals. 

Dear Dr. van Galen:

I'm pleased to inform you that your manuscript has been deemed suitable for publication in PLOS ONE. Congratulations! Your manuscript is now with our production department. 

Kind regards, 

on behalf of

Dr. Angel Abuelo 

Academic Editor

PLOS ONE